# Methyl and Ethyl Ethers of Glycerol as Potential Green Low-Melting Technical Fluids

**DOI:** 10.3390/molecules28227483

**Published:** 2023-11-08

**Authors:** Vadim Samoilov, Vladimir Lavrentev, Madina Sultanova, Dzhamalutdin Ramazanov, Andrey Kozhevnikov, Georgiy Shandryuk, Mariia Kniazeva, Anton Maximov

**Affiliations:** Topchiev Institute of Petrochemical Synthesis, Russian Academy of Sciences, 29 Leninsky Prospect, 119991 Moscow, Russia; lavrentev@ips.ac.ru (V.L.); sultanova@ips.ac.ru (M.S.); ramazanov@ips.ac.ru (D.R.); knyazeva@ips.ac.ru (M.K.); max@ips.ac.ru (A.M.)

**Keywords:** glycerol, ethers, antifreeze, heat transfer fluid, glycerol ether

## Abstract

The study is dedicated to the consideration of lower alkyl ethers of glycerol as potential components of low-melting technical fluids (e.g., heat transfer fluids, hydraulic fluids, aircraft de-icing fluids, etc.). Four isomeric mixtures of glycerol ethers (GMME—monomethyl; GDME—dimethyl; GMEE—monoethyl; GDEE—diethyl) were synthesized from epichlorohydrin and methanol/ethanol in the presence of sodium and subjected to detailed characterization as pure compounds and as aqueous solutions (30–90 vol%). The temperature and concentration dependencies of density, viscosity, cloud point, boiling range, specific heat capacity, thermal conductivity, and rubber swelling were obtained. On the basis of the data obtained, a comparison was made between the aqueous solutions of glycerol ethers and of other common bases for low-melting liquids (glycerol, ethylene glycol, and propylene glycol). Pure glycerol ethers could potentially be used as technical fluids in a very wide temperature range—from −114 to 150 °C. It was further demonstrated that in low temperature applications (e.g., in low-temperature chiller systems) the glycerol-ether-based aqueous heat transfer fluids could provide enhanced efficiency when compared to the glycerol- or propylene-glycol-based ones due to their lower viscosities and favorable environmental properties.

## 1. Introduction

The problem of bioglycerol valorization first became acute during the intensive growth of biodiesel fuel at the beginning of the 21st century. Despite the developed and successfully implemented industrial processes using bioglycerol as a chemical raw material (production of epichlorohydrin [1], production of biopropyleneglycol [2], and production of cyclic ketals [3]), glycerol retains its attractiveness as a raw material to produce valuable derivatives (Figure 1). This circumstance is dictated by the wide availability of glycerol and its low cost (determined by large volumes of biodiesel production), as well as its completely renewable nature. Moreover, the potential for the valorization (the generation of value) of a byproduct apparently influences the economic efficiency of the main production process. In this regard, the search for ways to valorize glycerol continues, which can be conditionally divided into two groups. The first group includes ways in which glycerol acts as an alternative raw material for the production of well-known and in-demand chemical products—for example, acrolein, which is now produced from propylene. The pathways of the second group are associated with the production of new derivatives based on glycerol, which can potentially replace existing solutions based on non-renewable or toxic products. Mostly considered as such derivatives are simple ethers, esters, and acetals/ketals of glycerol. The two main potential applications of these products are motor fuel components and green solvents. A striking example of this approach is the work on the study of glycerol ethers as green solvents, including their potential use as potential substituents for ethylene glycol ethers [4,5,6,7,8,9,10,11,12,13]. The vast majority of work on the production of fuel components is devoted to the production of *tert*-butyl, *sec*-butyl, or pentyl ethers of glycerol and their characterization as components of automotive fuels [14].

Another interesting potential application of lower alkyl ethers of glycerol is their use as cryoprotective solvents, described in [15,16]. It is clear that the main condition for such use is the safety (low toxicity) of the cryoprotective compound, which has also been studied in a number of recent studies [17,18,19]. In addition, their low-temperature properties, characterized by indicators of cloud point and pour point, are key. As follows from the cited works, 1-monomethyl and 1-monoethyl glycerol ethers have very low pour points (below −60 °C), and when mixed with water, they form solutions with a pour point below 0 °C. In combination with the observations made during the earlier work with glycerol ethers, this circumstance led the authors of the present study to the idea of the possibility of using these compounds in a different capacity—as components of low-solidifying technical liquids. This concept covers heat transfer fluids for use in transport and in stationary heating systems (including the so-called “antifreeze coolants”), working fluids for hydraulic drives of ground and air equipment, aircraft de-icing fluids, and other products for specialized applications. A common feature of these technical fluids is the requirement to maintain mobility and functionality at subzero temperatures, often combined with the requirement for high-performance properties at high temperatures. For example, an automotive coolant should remain liquid at low temperatures (commonly up to −30 °C) while having high heat transfer properties at the engine working temperatures (85–100 °C). Similar requirements apply to hydraulic fluids: they must remain mobile at low starting temperatures and avoid vapor formation at high temperatures due to the heating of the hydraulic system during operation.

Currently, the main substances used for the preparation of such liquids are ethylene glycol (EG), 1,2-propylene glycol (PG), and glycerol (Gly); in the case of aircraft de-icing fluids, diethylene glycol ethers are also used (diethylene glycol methyl ethers, diethylene glycol ethyl ether). Each of these compounds is characterized by a unique combination of physical properties that determines their advantages and disadvantages. In particular, EG, the main component of automotive antifreeze coolants, is widely available, inexpensive, and provides good viscosity characteristics of the resulting fluid. However, its main disadvantages are its toxicity and non-renewable nature, since the main source of this product is the ethylene oxidation process. Compared to EG, PG and Gly are green alternatives because these compounds are non-toxic and can be produced from renewable raw materials. However, aqueous solutions of these compounds have significantly higher viscosity and pour points compared to EG-based solutions, limiting their use. In addition, none of these substances can serve as the basis for a working fluid that retains mobility at temperatures below −65 °C.

The limited literature on the physical properties of glycerol methyl and ethyl has hindered an assessment of the potential of these compounds as components of low-solidification technical liquids. Lower glycerol ethers can be obtained in a one-stage esterification reaction of glycerol with methanol [20,21] and ethanol [22,23,24,25,26,27,28,29]. This offers the possibility of obtaining these products entirely from renewable raw materials. In principle, the reaction of glycerol with a simple alcohol can result in two isomeric monoethers, two isomeric diethers, and one triether. In the present work, methyl and ethyl ethers of glycerol were investigated as mixtures of isomeric compounds (Figure 1), the composition of each of which was determined by the selected synthesis method.

A study aimed at considering methyl and ethyl ethers of glycerol as potential components of technical fluids was conducted. Various technical fluids share similar criteria for evaluating their performance properties. For example, in both brake fluids and coolants, the lower temperature limit of application is determined by the temperature at which pumpability is lost. Therefore, to evaluate the possibilities of using glycerol ethers, it is necessary to have data on the relevant physical properties of these compounds. To assess the potential of these compounds, samples (isomeric mixtures) of methyl and ethyl mono- and diethers of glycerol (GMME—monomethyl; GDME—dimethyl; GMEE—monoethyl; GDEE—diethyl) were synthesized. The obtained samples were subjected to “general” characterization as pure substances (describing the main properties) and “special” characterization in the form of aqueous solutions allowing the application potential to be estimated. This involved obtaining data on temperature and concentration dependencies of changes in density, kinematic viscosity, cloud point, and other relevant physical properties. Based on the data acquired, prototypes of the low-melting heat transfer fluids were prepared and characterized. The results obtained were compared with the literature data for conventional heat transfer fluids prepared with EG, PG, or Gly.

## 2. Results

The synthetic glycerol ether sample formulations tested in this paper are presented in Table 1 (the GC analysis results are given in Appendix A). A glycerol esterification reaction catalyzed by a heterogeneous acid catalyst (Amberlyst 36) was used to prepare monoethers samples (GMME and GMEE). As known from the literature, such synthesis results in an isomeric mixture of monoethers, in which the ratio of 1-monoether/2-monoether is about 50:50 [30], while higher selectivity to 1-monoether is typical for alkali-mediated synthesis from epichlorohydrin [31]. Samples of diethers were obtained by synthesis from epichlorohydrin in the base medium. Typically, synthesis under such conditions tends to exhibit increased symmetric diether selectivity [6]. As shown by the obtained results (Table 1), the synthesis from epichlorohydrin in the presence of sodium alcoholate proceeded with high symmetric ether regioselectivity. For the synthesis of monoethers, a direct acid-catalytic esterification reaction (typically running with regioselectivity of about 10–20% to 2-monoether) was used. However, the isomeric composition of each sample was also influenced by the purification method used. For instance, the monoether mixtures were enriched with 1-monoethers through fractional distillation.

As is known, the upper temperature limit of service fluid operation in the first approximation is determined by its boiling point and saturated vapor pressure at elevated temperatures. Experimental data on the dependence of the boiling point on pressure (Table 2) show that the boiling points of glycerol monoethers (GMME—220–225 °C; GMEE—222–225 °C) are lower than the boiling point of glycerol (290 °C) and higher than the boiling point of PG (187 °C). The lowest boiling point (164–167 °C) among the samples obtained was recorded for GDME; this value is quite close to the reported earlier [32,33]. The same concordance was observed for the GDEE: the boiling point previously reported by Leal-Duaso et al. was 188 °C, which is 4–7 °C lower than the value obtained by us. The obtained data can be used to approximate the pressure of saturated vapors of ethers at different temperatures.

The lower temperature limit of service fluid operation is determined by the ability to maintain mobility (pumpability) at low temperatures. The first parameter characterizing this property is the freezing point. It should be noted that the standard method, ASTM D1177, that is used to control the low-temperature properties of water-based coolants, defines the freezing point as the temperature at which solid-phase crystallization from the solution is observed, detected by the constant temperature of the solution with a continuing decrease in the temperature of the cooling bath. Based on this, in the present study, the obtained samples of glycerol ethers were characterized by the freezing point indicator using the same method but with optical detection of the crystallization point by the appearance of the first crystals in the cooled sample.

All the pure samples of the ethers obtained had pour points below −60 °C, which is why it was impossible to measure the exact values using the available equipment (Table 3). In this regard, crystallization temperatures for pure compounds were estimated from the phase transition temperature recorded by differential scanning calorimetry. The studied isomeric mixtures of glycerol ethers have extremely low pour points. The values for dialkyl ethers (−114 and −105 °C) are close to the pour point of ethanol (−114 °C), while the values for monoalkyl ethers (−90 and −94 °C) are similar to the pour point of methanol (−98 °C). These values are significantly lower than the melting points of such common glycol ethers as diglyme (−64° C), 2-ethoxyethanol (−70 °C), dipropyleneglycol methyl ether (−80 °C), and diethylene glycol monomethyl ether (−84 °C).

The most effective decrease in the crystallization temperature in aqueous solutions was observed for monoethers samples: a crystallization temperature below −60 °C was achieved at a concentration of GMME or GMEE of about 70 wt% (Table 3). For GDME, this freezing point required a higher ether concentration (at least 80 wt%). The lowest efficiency in reducing the freezing point was observed for GDEE: a 90 wt% aqueous solution of this compound had a crystallization start temperature of exactly −60 °C. Thus, GDEE is a clear “outsider” among the studied glycerol ether derivatives in terms of the effectiveness of reducing the freezing point of the aqueous solution.

The available data allow for an initial comparison between glycerol ethers, pure glycerol, and PG. The last two compounds are widely used as components of low-solidification technical fluids (both hydraulic and heat transfer fluids). To achieve a working fluid with a freezing point not higher than −15 °C, the required concentration of organic matter is approximately the same for Gly, PG, GMME, GMEE, and GDME (about 40 wt%). The minimum concentration of PG to produce a liquid with an operating temperature of up to −40 °C is about 55 wt%, while in the case of glycerol, the concentration should be increased to 65 wt%. All glycerol ethers except GDEE occupy an intermediate position between PG and Gly, providing a freezing point below −40 °C at a concentration of 60–65 wt%. Of particular interest is the task of obtaining a liquid with a freezing point below −60 °C. According to the literature data, it is impossible to achieve freezing point values below −45 °C using glycerol [35]. Aqueous solutions of PG with a concentration of more than 60 wt% have a pour point below −70 °C [34] and the concentration corresponding to a freezing point of −60 °C falls within the range of 60–65 wt%. Similar values were observed for glycerol monoethers: the crystallization temperatures of 65 wt% of GMME and GMEE solutions were −55 and −58 °C, respectively. In the case of GDME, crystallization temperatures below −60 °C were typical for solutions with concentrations above 70 wt%. Therefore, the obtained ethers are generally comparable to PG and glycerol in their ability to reduce the crystallization temperature of aqueous solutions.

The dependencies of density variation on concentration obtained for aqueous solutions of glycerol ethers exhibit a nonlinear character (Figure 2). For GMME, GMEE, and GDME samples, the dependence was nearly linear in the concentration range of 0–60 wt%, losing its linearity at higher concentrations. This correlates with the observed changes in freezing points, where a sharp decrease in the parameter for the samples is noted at concentrations above 60 wt%. For the GDEE sample, the dependence had a nonlinear character over the entire concentration range of aqueous solutions. This behavior (the so-called ‘contraction’ of the solution) is typical for aqueous solutions where the formation of strong hydrogen bonds takes place—for instance, with aqueous mixtures of glycerol [35] and propylene glycol [34].

Similar dependencies were obtained for temperatures of 10 and 40 °C (Appendix A). This made it possible to calculate the coefficients of linear dependencies of density on temperature in the range of 10–40 °C. The temperature dependence is expressed by a function of the form:(1)$$\rho \left(T\right)=k\times T+\rho_{0}$$
where *ρ*(*T*) is the density (kg m^−3^) at a temperature of *T* (°C), *k* is the slope (kg m^−3^ °C^−1^), and *ρ*_0_ is the calculated density at a temperature of 0 °C (kg m^−3^).

The coefficients *k* and *ρ*_0_ calculated as a result of the approximation of the experimental data are presented in Table 4. After examining the temperature dependencies of density for aqueous solutions of glycerol and PG [34,35], it was determined that the linearity of this dependence was usually maintained at least in the temperature range from −20 to 80 °C. Therefore, the equations obtained for the temperature dependencies of density were subsequently used for linear extrapolation of density values at temperatures of −20 and 80 °C. These temperatures were selected as extreme points for comparing the physical properties of aqueous solutions of ethers.

The viscosities of aqueous solutions of glycerol ethers were measured using a capillary viscometer (Table 5). One of the reasons for choosing this method is the ability to reliably control the accuracy of measurements. The representativeness of the obtained values is supported by both the metrological correctness of the measurements and the values obtained for pure water (Appendix A). As expected, the viscosities of monoethers are significantly higher than those of diethers; the viscosities of ethyl-substituted compounds are slightly higher than the viscosities of methyl ethers. The viscosity values of pure monoethers at 20 °C (49.0 and 50.6 mm^2^ s^−1^ for GMME and GMEE, respectively) are close to those for pure PG (54.0 mm^2^ s^−1^) and significantly lower than the viscosity of pure glycerol (1117.9 mm^2^ s^−1^). The viscosities of pure diethers (4.2 mm^2^ s^−1^) are almost an order of magnitude lower than the viscosity of PG and approach in their values to aliphatic alcohols—isopropanol (3.0 mm^2^ s^−1^), n-butanol (3.6 mm^2^ s^−1^), and isoamyl alcohol (5.3 mm^2^ s^−1^).

The results obtained demonstrate a significant difference in the behavior of aqueous solutions of glycerol mono- and diethers. Both glycerol monoethers (GMME and GMEE) were characterized by the following behavior: the viscosity of the aqueous solution decreased as the water concentration increased, both at positive temperatures and at −20 °C. This behavior is characteristic, in particular, of aqueous solutions of PG and glycerol [34,35]. At the same time, the viscosities of all the 80 wt % diether solutions (GDME and GDEE) were higher than the viscosities of pure ethers and with a further increase in the concentration of water, viscosity decreases were observed. Therefore, the dependence of viscosity on concentration for aqueous solutions of these compounds is extremal, with a maximum in the region of 60–100%. In combination with the observed nonlinearity of the concentration dependencies of density and crystallization temperature, this indicates the existence of an aqueous solution in which the configuration of water clusters has minimal energy. This interesting question has not been explored separately, since it falls beyond the scope of this work.

The possibility of using aqueous solutions of organic substances of a certain quality depends, among other things, on the compatibility of the substance with wetted construction materials. For glycerol ethers, there is no expectation of significant corrosive activity toward metals, as these compounds combine the functional groups of alcohols and ethers. Additionally, the substitution of hydroxyl groups of polyols leads to the production of ethers (alkoxy ethers), which exhibit lower polarity and high soluble properties with respect to polymers. Therefore, to assess the potential for the use of glycerol ethers, it is necessary to have at least preliminary information about their effect on the polymer materials, particularly seals. In the standard GOST 9.030 test used to assess the quality of water-based coolants, the swelling of the rubber sample after exposure to the liquid should not exceed 5 wt % (swelling is determined by the increase in the mass of the dried sample of swollen rubber compared to the mass before swelling). As can be seen from the data in Table 6, the weight gain of the rubber sample did not exceed 5 wt % for all studied aqueous solutions of ethers (GMME, GDME, and GMEE). Therefore, aqueous solutions of glycerol ethers can be considered as having satisfactory compatibility with oil-resistant rubber.

By analyzing the obtained data on density, viscosity, and crystallization temperature by thermophysical properties, it is possible to assess the effectiveness of using glycerol ethers as heat carriers. To perform this process, the compositions of prototype solutions of technical liquids for two groups were selected. Prototypes of the first group provide performance at minimum temperature of −60 °C: these are formulations GMME_80_, GDME_80_, and GMEE_80_ (the index at the name indicates the weight concentration of ether in an aqueous solution). Prototypes of the second group (GMME_60_ and GDME_60_) provide performance at minimum temperatures up to −40° C. For prototype solutions, data on specific heat (DSC method), thermal conductivity coefficient (calculation method), initial boiling point (ASTM D86), and flash point (ASTM D56) were additionally obtained (DSC curves could be found in Appendix A). These indicators provide insights into the quality and conditions for the use of these technical fluids. For comparison, similar formulations based on PG and Gly that provide the same freeze protection (PG_80_ below −60 °C; PG_60_ and Gly_70_ at −40 °C) were selected. The obtained data are presented in Table 7.

The first part of the table provides a comparison of properties (including thermophysical) for pure substances: GMME, GDME, GMEE, PG, and Gly. The heat capacity and heat conductivity values of glycerol ethers are comparable to those for PG and glycerol. However, the flash point values for glycerol ethers stand out: dimethyl ether, with a flash point of only 49 °C, is a flammable liquid. For monoethers, the flash point values (GMME—133 °C, GMEE—145 °C) determined by us (closed cup method) were significantly higher than the values for PG (99 °C). According to the literature, PG solutions with water concentration > 15 wt % do not flash; glycerol solutions do not flash as well, until water evaporates to reach a glycerol concentration of 97.5 wt %. A similar behavior was observed for aqueous solutions of glycerol ethers. For example, when measuring the flash point for the GDME_80_ solution at temperatures up to 100 °C, no flash was observed. With a further increase in temperature, water boiled in the crucible, after which a flash was achieved at 111 °C. As the temperature further increased to 113 °C, an attempt to flash led to a steady burning of the test ether in the crucible. Therefore, the addition of 20 wt % water is sufficient to “suppress” the flammability of even the most volatile of the tested samples. For other samples of ether solutions (GMME_80_, GMEE_80_, GMME_60_, and GDME_60_), flash at temperatures up to 100 °C was also absent.

In addition, when comparing pure substances, viscosity values at T = −20 °C are noteworthy: for all glycerol ethers, this value is significantly lower than for PG and Gly. Thus, pure GMME, GDME, and GMEE have an advantage over PG and Gly with regard to the lower temperature limit of application. Their extremely low pour points combined with relatively low viscosity values make it possible to use them at least at temperatures down to −90 °C.

The second part of the table contains data on prototype liquids designed for the lowest operating temperature of −60 °C. Such coolants and hydraulic fluids may find demand in conditions with extremely low starting temperatures, such as in air technology, in the Arctic or Antarctic. This challenge cannot be addressed using aqueous solutions of glycerol, as the minimum crystallization temperature for them is about −44 °C (for a solution of Gly_66.7_). In the case of aqueous PG solutions, the required minimum concentration of glycol in water is about 65 wt % [34]. However, for comparison purposes, all bases of low-solidified liquids were taken at the same concentration of 80 wt %. The thermophysical properties of all four solutions (PG_80_, GMME_80_, GDME_80_, and GMEE_80_) are at the same level. Dynamic viscosities at T = 80 °C have similar values for PG (2.0 mPa·s) and monoethers (2.5 mPa·s), while the viscosity of the GDME_80_ solution is noticeably lower (1.2 mPa·s). However, at a temperature of −20 °C, the PG_80_ solution has the viscosity value of 400 mPa·s, while the value for the GDME_80_ sample was six times lower (66 mPa·s). Monoether solutions exhibited viscosity–temperature and thermophysical properties close those of the PG_80_ solution, except for viscosity at negative temperatures. The GDME_80_ solution, with similar thermophysical properties, had significantly lower viscosity at both positive and negative temperatures. Low viscosity can be considered an advantage for a heat transfer fluid, reducing energy consumption for pumping and intensifying heat transfer.

Prototype liquids of the third group are designed for an operating temperature of up to −40 °C. In the case of PG, GMME, and GDME, the problem is resolved by an organic compound concentration of 60 wt %, while glycerol requires a concentration of about 67–70 wt %. According to the data obtained, all four solutions have comparable values for heat capacity and thermal conductivity and mainly differ in viscosity. The low-temperature viscosity of glycerol ether solutions was significantly lower than that of PG and Gly solutions. At T = 80 °C, this difference is significant primarily when comparing glycerol with other solutions.

Of note are the boiling points of aqueous solutions of glycerol ethers. These solutions did not exhibit the typical pattern of an increase in the boiling point of an aqueous solution with a decrease in the water content. Instead, the boiling of all tested mixtures began in the range of 100–101 °C, accompanied by a sharp increase in temperature only when water had been completely evaporated (Table 8).

Hence, the use of glycerol ethers as bases for the production of aqueous low-solidifying liquids instead of glycerol and PG appears most viable in scenarios where very low operating temperatures are required, especially below −40 °C, and particularly below −60 °C. Pure glycerol monoethers can be used when the working fluid needs to maintain mobility at temperatures below −60 °C while maintaining satisfactory properties at elevated operating temperatures (for example, 80 °C).

For a liquid flowing in turbulent mode inside of a pipe (internal diameter, *d*) with a mass flux *j*, the heat transfer coefficient could be calculated using the Dittus–Bölter equation (Equation (1)):(2)$$\frac{hd}{\lambda }=0.023{\left(\frac{jd}{\mu }\right)}^{0.8}{\left(\frac{\mu C_{P}}{\lambda }\right)}^{0.4}$$
where *h* is the heat transfer coefficient and *λ*, *μ*, and *C_P_* are the thermal conductivity coefficient, the dynamic viscosity, and the specific heat of the liquid, respectively. The equation is known as a correlation affording the accuracy of calculation of about ±15%. Hence, the calculations performed in the present study do not pretend to be precise, yet they allow the qualitative comparative evaluation to be conducted.

Let heat transfer coefficients h should be compared for two heat transfer cases differing from each other only in the thermophysical properties of the liquid flowing (the pipe diameter *d* and the liquid mass flux *j* are equal). The expression (Equation (2)) could be then transformed as follows:(3)$$\frac{h_{1}}{h_{2}}={\left(\frac{C_{P1}\times \mu_{2}}{\mu_{1}\times C_{P2}}\right)}^{0.4}\times {\left(\frac{\lambda_{1}}{\lambda_{2}}\right)}^{0.6}$$

By solving the (Equation (3)) for different pairs of the heat transfer fluid prototypes (taken from Table 7), one could compare the heat transfer coefficients. For example, in the case where the liquid temperature is 80 °C, substitution of PG_80_ to GDME_80_ augments the heat transfer coefficient by 8%. The viscosity difference between the PG- and GDME-based solutions rises with temperature decrease: at 80 °C PG_60_ and GMDE_60_ mixtures had dynamic viscosities of 1.3 and 1.1 mPa*s, respectively, whereas at −20 °C these amounted to 60 and 140 mPa*s, respectively. Thus, at −20 °C the use of GDME_60_ heat transfer fluid leads to the *h*_1_/*h*_2_ = 1.22 (+22%). Moreover, the specific heat (*C_P_*) values obtained by us through DSC measurements look somewhat lowered (by about 8%) if compared with, e.g., the values reported for GDME by Andreeva et al. [36]. If for the latter case a recalculation is made using the *C_P_* = 3.25 J mol^−1^ K^−1^ (instead of 3.02 J mol^−1^ K^−1^ measured by us), then the *h*_1_/*h*_2_ makes 1.26 (+26%). The obtained difference derives from the lower viscosity of the glycerol-ether-based heat transfer fluid compared to the PG-based one. Further temperature lowering multiplies the effect: at −40 °C, the *η*_−40_ values for PG_80_ and GDME_80_ amount to 4100 [34] and 365 (data by the authors) mPa*s, respectively. Here, it is obvious that in this case the difference in heat flows will be dramatic. Moreover, 80 wt % aqueous solution of ethylene glycol has freezing point of −47 °C and *η*_−40_ = 449 mPa*s, comparable to that of the GDME_80_ solution (<-60 °C and 365 mPa*s). The viscosity difference was much lower for the PG_60_/GMME_60_ pair, so in this case the heat transfer coefficient surplus made only 4.5%. Apparently, this difference should be much greater if the glycerol-ether-based mixtures are compared with the glycerol-based ones: at −20 °C, GDME_60_ coolant prototype (*η*_−20_ = 60 mPa*s) had 87% higher *h*-coefficient than the Gly_70_ solution (*η*_−20_ = 394 mPa*s).

To conclude, glycerol-ether-based aqueous mixtures could be potentially used as heat transfer fluids in low-temperature applications, where subzero and especially extremely low operation temperatures are required (for instance, in low-temperature refrigeration systems). In these cases, the use of these compounds could be advantageous in comparison to the PG- or Gly-based analogues due to the considerably lower viscosity intensifying heat flow and reducing energy consumption for the liquid circulation pumping. Moreover, earlier reports indicate that the methyl and ethyl ethers of glycerol have favorable ecotoxicity properties [17,18] allowing application of these compounds in direct contact with live cells [37,38]. These properties are important in the scope of the Green Chemistry’s Principle #4 “Designing safer chemicals”: as it has been shown, glycerol-based solutions have poor performance properties compared to the glycerol-ethers-based ones. Propylene glycol solutions indicate comparable performance at temperatures above ≈−20 °C, while performing less efficiently below this point. The other existing analogues, ethylene glycol and its derivatives, have good performance properties yet being known as fossil-based and toxic compounds [39]. Thus, the use of bio-based glycerol ethers in those applications where low operation temperatures are required could be advantageous in terms of energy efficiency (due to higher heat flux values) and safety both for the environment and for people.

## 3. Materials and Methods

### 3.1. Reagents

The materials used for synthetic purposes are listed in Table 9. The commercially available materials were used without any further purification.

### 3.2. Synthesis of the Glycerol Alkyl Ethers

#### Synthesis of the 1,3-Diethyl Ether of Glycerol (GDEE)

A double-necked round-bottom flask equipped with a reflux condenser, dropping funnel and a magnetic stirrer was charged with ethanol (902.9 g, 20 mol). Then, metallic sodium (143.8 g, 6.3 mol) was carefully added as thin flattened chips. After the complete dissolution of sodium, the flask was heated to 45–50 °C and epichlorohydrine (133.0 g, 1.4 mol) was added dropwise through the dropping funnel. As epichlorohydrine was introduced, the reaction continued for 1 h with stirring. As the reaction ended, the reaction mixture was filtered to separate the NaCl formed; the ethanol was separated with a rotavap (residual pressure 20 mmHg, bath temperature 50 °C). The residue was separated using vacuum rectification (residual pressure 20 mmHg, reflux ratio 3–4) to yield the isomeric mixture of glycerol diethyl ethers (178 g) recovered at 86–89 °C (20 mmHg). The 1,3-diethyl ether was the main constituent of the mixture.

1,3-di-ethoxypropan-2-ol, 178 g, 86%. bp 192–195 °C. Found (%): C, 56.64; H, 10.85. Calculated for C_7_H_16_O_3_ (%): C, 56.73; H, 10.88. The GC purity was 98.6%. The MS and H^1^-NMR spectral data appropriately corroborated with the NIST database and with the previously reported results of [6].

### 3.3. Synthesis of the 1,3-Dimethyl Ether of Glycerol (GDME)

A double-necked round-bottom flask equipped with a reflux condenser, dropping funnel, and a magnetic stirrer was charged with methanol (2082.6 g, 65 mol) and sodium hydroxide (244 g, 6.3 mol). After the complete dissolution of the alkali, the flask was heated to 45–50 °C and epichlorohydrine (413.3 g, 4.4 mol) was added dropwise through the dropping funnel. As epichlorohydrine was introduced, the reaction continued for 1 h with stirring. As the reaction ended, the reaction mixture was filtered to separate the NaCl formed; the methanol was separated with a rotavap (residual pressure 20 mmHg, bath temperature 50 °C). The residue was separated by vacuum rectification (residual pressure 20 mmHg, reflux ratio 3–4) to yield the isomeric mixture of glycerol dimethyl ethers (423 g) recovered at 68–82 °C (20 mmHg). The 1,3-dimethyl ether was the main constituent of the mixture.

1,3-di-methoxypropan-2-ol, 423 g, 81%. bp 164–167 °C. Found (%): C, 50.07; H, 10.12. Calculated for C_5_H_12_O_3_ (%): C, 49.98; H, 10.07. The GC purity was 99.2%. The MS and H^1^-NMR spectral data appropriately corroborated the NIST database and the previously reported results of [6].

### 3.4. Synthesis of the Glycerol Monoethyl Ether (GMEE)

The monoethyl ether of glycerol was obtained by the direct etherification of glycerol with ethyl alcohol. In a stainless-steel batch stirred reactor (internal volume, 1 L) a mixture of glycerol and ethanol (1:1 mol, 600 mL) was loaded with a catalyst charge (Amberlyst 36, 5 wt % to glycerol). The reactor was twice purged and finally pressurized (20 bar) with nitrogen. The reaction was carried out at T = 160 °C, at an autogenerated pressure for 24 h. The cooled reaction mixture was filtered from the catalyst; the excess alcohol was separated using a rotavap (residual pressure 20 mmHg, bath temperature 50 °C). The residue was separated through vacuum rectification (residual pressure 20 mmHg, reflux ratio 3–4) to yield the isomeric mixture of glycerol monoethyl ethers recovered at 121–126 °C (20 mmHg). The 1-monomethyl ether was the main constituent of the mixture. The average yield of the purified ether was 27%.

3-ethoxypropane-1,2-diol, bp 222–225 °C. Found (%): C, 49.87; H, 10.09. Calculated for C_5_H_12_O_3_ (%): C, 49.98; H, 10.07. The GC purity was 97.3%. The MS spectral data appropriately corroborated the NIST database.

### 3.5. Synthesis of the Glycerol Monomethyl Ether (GMME)

The monomethyl ethers of glycerol were prepared and isolated according to the method reported previously by us [20].

3-methoxypropane-1,2-diol, bp 220–225 °C. Found (%): C, 45.37; H, 9.43. Calculated for C_4_H_10_O_3_ (%): C, 45.27; H, 9.50. The GC purity was 95.9%. The MS spectral data appropriately corroborated the NIST database.

The analysis of the isomeric composition and of the purity was accomplished with GC-FID (Kristallyuks-4000M apparatus (Meta-Khrom, Yoshkar-Ola, Russia), Supelco Nukol capillary GC column (30 × 0.25 × 0.25) (Merck, Darmstadt, Germany), helium carrier gas.

### 3.6. Physical Properties Determination

The standard methods used for the characterization of hydrocarbon mixtures are listed in Table 10.

### 3.7. Boiling Point at Reduced Pressure

The ‘boiling point—pressure’ dependences were obtained by the vacuum rectification of the compounds under different residual pressures using the laboratory rectification column (random glass ring packing, 40 cm column length, reflux ratio ∞). The desired operation pressure measured at the column head was set by regulating the performance of the vacuum system. The system was allowed to become stabilized over 15–20 min, after what the mean boiling temperature was recorded three times every 5 min. After the successful measurement, the residual pressure was changed, and the next measurement was carried out.

### 3.8. Rubber Swelling Test

The rubber swelling test was carried out in accordance with the GOST 9.030-74 “Unified systems of corrosion abd ageing protection. Vulcanized rubbers. Method of testing resistance to attack by corrosive media in limp state” (method B). Oil- and fuel-resistant nitrile butadiene rubber (resin mixture No. 57-5006, produced by OZ RTI, Podolsk, Russian Federation) was used as the test sample. Weighed pieces of the resin were immersed in the test liquids (aqueous solutions of glycerol ethers) and held at 100 ± 2 °C for 70 h, then dried until constant weight of the swelled sample was reached. The relative weight surplus of the rubber sample is the swelling degree expressed in wt %. Two parallel runs were performed for each measurement.

### 3.9. Specific Heat Capacity

The specific heat measurements (between −40 and 100 °C) were conducted using DSC3+ (Mettler Toledo, Switzerland), with a main sensor CERAMIC:HSS9+ HIGH and a standard sapphire etalon.

### 3.10. Thermal Conductivity Coefficient

Thermal conductivity of pure isomeric mixtures was calculated using Sato-Riedel equation [40]. Deviations of 20% can be expected but as usual the errors should not exceed 15% [41].
$$\lambda =\frac{1.1053\times (3+20\times {\left(1-T_{r}\right)}^{2/3})}{\sqrt{M}\times (3+20\times {\left(1-T_{br}\right)}^{2/3})};\;T_{r}=\frac{T}{T_{c}};\;T_{br}=\frac{T_{b}}{T_{c}},$$
where λ—liquid thermal conductivity of the pure component; M—molecular weight; $T_{b}$—boiling point of the pure component; $T_{c}$—critical temperature of the pure component.

Thermal conductivity of liquid aqueous mixtures was calculated using Li mixing rule [40]. According to Ref. [42], this one is recommended for aqueous mixtures at atmospheric pressure but maximum deviation of 33% was obtained.
$$\lambda_{mix}=\sum_{i}\sum_{j}\varphi_{i}\varphi_{j}\lambda_{i,j};\;\lambda_{i,j}=\frac{2}{\frac{1}{\lambda_{i}}+\frac{1}{\lambda_{j}}};\;\varphi_{i}=\frac{X_{i}V_{m,\;i}}{\sum_{j}^{n}X_{j}V_{m,j}}\;$$
where $\lambda_{mix}$—thermal conductivity of liquid mixture; $\lambda_{i}$—liquid thermal conductivity of pure component; $X_{i}$—mole fraction of pure component in mixture; $V_{m,i}$—molar volume of pure component.

## 4. Conclusions

Four samples of glycerol alkyl ethers (monomethyl—GMME; monoethyl—GMEE; dimethyl—GDME; diethyl—GDEE) were synthesized and purified. The pure compounds were subjected to a detailed characterization of their physical properties including density and kinematic viscosity at different temperatures: boiling point at reduced pressure and freezing point. It has been shown that pure ethers exhibit a wide range of liquid states, with freezing temperatures below −90 °C and boiling points above 164 °C (diethers) and 220 °C (monoethers). These four ethers were also considered as bases for the preparation of aqueous blends with potential applications as low-melting technical fluids such as de-icing fluids, hydraulic fluids, and heat transfer fluids for stationary and automotive applications. Five prototypes of glycerol-ether-based antifreeze fluids were prepared: GMME_80_, GDME_80_, GMEE_80_ (80 wt% ether content), GDME_60_, and GMME_60_ (60 wt% ether content). The physical and thermophysical properties of these fluids were determined and compared with the properties of conventional PG and glycerol-based mixtures. For applications requiring freeze protection below −60 °C, the performance of the 80 wt% monoether solutions is fully comparable to that of the 80 wt% PG solution (PG_80_). Moreover, the viscosity of the GDME80 sample was significantly lower than that of the PG_80_ at both low and high temperatures (−20 and 80 °C), suggesting potential for increased heat transfer. Among the mixtures providing antifreeze protection at −40 °C (PG_60_, GDME_60_, GMME_60_, and Gly_70_), the glycerol-based fluid had the highest viscosity. In contrast, the viscosities of the ether-based mixtures were comparable to those of PG_60_. Although the viscosity of the GDME_60_ at −20 °C was again lower (55.2 vs. 140.0 mm^2^ s^−1^), at 80 °C, the difference was negligible (1.1 and 1.3 mm^2^ s^−1^). Additionally, while the ether-based blends offer lower viscosity along with favorable environmental properties such as a potentially renewable origin and low toxicity, they were found to have poor boiling points compared to the PG_80_ blend, which had a boiling point of 117 °C, whereas all the 80 wt% glycerol ether solutions had boiling points of only 101 °C.

## Data Availability

Selected datasets (incl. viscosity, density and DSC measurements, raw data, uncertainty analysis) could be found in the Appendix A section.

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
