# Peer review of "Methyl and Ethyl Ethers of Glycerol as Potential Green Low-Melting Technical Fluids"

_molecules, 2023, doi:10.3390/molecules28227483_

Round 1

Reviewer 1 Report

Comments and Suggestions for Authors

Reviewer comments:

Abstract

 Basics of Synthesis processes of glycerol ethers from epichlorohydrin

 Make it easy to digest and relate the sentences with the used terms.

 provide insights into the specific applications in which glycerol ether-based aqueous heat transfer fluids show enhanced efficiency over glycerol- or propylene glycol-based alternatives?

 Are there any notable environmental or safety advantages associated with using glycerol ethers as low-melting technical fluids compared to traditional options?

Introduction:

 Give few knowledge about volarization history,background and importance in industry and present role in production of various different chemicals

https://doi.org/10.1016/j.electacta.2023.142383

https://doi.org/10.1111/ijfs.16443

 Can also discuss properties and characteristics determining the performance of fluids such as bakelite coolant and relate them with lower temperature behavior https://doi.org/10.1016/j.ijheatmasstransfer.2017.01.094

 Physical properties of all four components should be explained concisely

Results:

 Can you elaborate on why the selectivity varies between 1-monoether and 2- monoether in the synthesis of monoethers using acid catalysis versus alkali-mediated synthesis from epichlorohydrin? Additionally, what are the contributing factors to this difference in selectivity?

https://doi.org/10.1002/anie.202201739

 What is the reason behind the preference for symmetric diethers and how does the base medium affect the selectivity of diethers?

 literature review about the Moveability of temperature with variations of fluids

 there is no literature work on the comparison of the physical values of temperature

and pressure (table 2)

https://doi.org/10.1016/j.fuproc.2023.107908

https://doi.org/10.1021/acs.biomac.2c00223

Material and method

 Table 10 should have all values compared with the literature because it has all standard values (modify the table 10)

https://doi.org/10.1039/D2PY00551D

https://doi.org/10.1021/acsomega.1c06917

Conclusion:

 Elaborate these environmental properties and give reason why they are important in the context of antifreeze fluids and other potential applications?

 how do these properties relate to their potential applications in de-icing fluids, hydraulic fluids, and heat transfer fluids?

Comments on the Quality of English Language

Authors need to look for revised version

Reviewer 2 Report

Comments and Suggestions for Authors

The paper considered presents a novel route to synthesis of lower alkyl ethers of glycerol. It is shown that these products can be potentially used as low-melting technical fluids (e.g. heat transfer fluids, hydraulic fluids, aircraft deicing fluids, etc.) in a very wide temperature range – from - 114 to 150 °C. Paper contains experimental information about synthesis of four isomeric mixtures of glycerol ethers (GMME – monomethyl, GDME – dimethyl, GMEE – monoethyl, GDEE – diethyl) from epichlorohydrin. As pure compounds and as aqueous solutions (30–90 vol%)  were subjected to detailed characterization  (the temperature and concentration dependencies of density, viscosity, cloud point, boiling range, specific heat capacity, thermal conductivity, and resin corrosion). Moreover, additional characterization data are presented as supplementary ones for understanding. The positive comparison was made between the physical properties of aqueous solutions of glycerol ethers and conventional compounds  (glycerol, ethylene glycol, propylene glycol). My comments are:

1. It would be useful to explain the nonlinear change in density with the composition of the water-organic mixture (Fig.2)

2. It would be useful to indicate in the Abstract a specific example of glycerol ether for potential practical use, and not only indicate the general temperature range.

In whole, the paper meets the requirements of the journal and can be published after explaining my comments (Minor Revision).
